# A patient-centred care model for patients with complicated multimorbidity: Protocol for a pilot cluster randomised trial in general practice, municipalities, and hospitals

**Sanne Lykke Lundstrøm**[1,2]*, **Nina Kamstrup-Larsen**[1,3], **Barbara Ann Barrett**[1,3], **Louise Marie Bidstrup Jørgensen**[3], **Solvej Skriver Hansen**[3], **John Sahl Andersen**[3], **Bolette Friderichsen**[4], **Anders Stockmarr**[5], **Anne Frølich**[1,3]

1 Innovation and Research Centre for Multimorbidity, Slagelse Hospital, Region Zealand, Slagelse, Denmark, 2 Center for Clinical Research and Prevention, Bispebjerg and Frederiksberg Hospital, Frederiksberg, Denmark, 3 Department of Public Health, Research Unit for General Practice and Section of General Practice, University of Copenhagen, Copenhagen, Denmark, 4 The Danish College of General Practitioners, Copenhagen, Denmark, 5 Department of Applied Mathematics and Computer Science, Section for Statistics and Data Analysis, Technical University of Denmark, Copenhagen, Denmark

* Sanne.lykke.lundstroem@regionh.dk

**Data Availability Statement:** No datasets were generated or analysed during the current study.

## Abstract

### Introduction

Current care is inadequate for patients with complicated multimorbidity, and frequently results in fragmented care. There is no widely agreed-upon optimal organisation of health-care services for this patient group. By drawing upon existing literature and prior studies, we developed a patient-centred complex intervention for multimorbidity (CIM) and subsequently refined it into CIM version 2 (CIM2). This paper describes the study protocol for a pilot cluster randomised control trail (RCT) evaluating the effectiveness of a general practice-based intervention.

### Methods

CIM2 aims to support integrated care for patients with complicated multimorbidity. CIM2 comprises five elements: 1) Training healthcare professionals, 2) an extended overview consultation in general practice, 3) a nurse care coordinator in general practice supporting the planning of the patient trajectory, 4) follow-up care services in general practice, and 5) improving the integration of care between general practice, municipality, and hospital. The pilot cluster RCT involve 350 patients with complicated multimorbidity across 14 general practices in Region Zealand and The Capital Region of Denmark. Patients are randomly assigned to either the intervention group or the usual care group. The primary outcome measure is the patients experience of quality of care measured by the Patient Assessment Chronic Illness Care Questionnaire (PACIC). Secondary outcomes include the patient's health-related quality of life, measured by the EuroQol-5 Domain questionnaire (EQ-5D-5L) and the treatment burden measured by the Multimorbidity Treatment Burden Questionnaire

Data in the pilot study cannot be shared publicly because of restrictions on data availability according to Danish registration. The participants in the study have not given consent to make these data publicly available. Authorized scientific institutions can request access to this data from Statistics Denmark for individual researchers within or outside the country. Requests should be directed to Statistics Denmark at: https://www.dst.dk/en.

**Funding:** This work was supported by: Den forskningsfremmende pulje i Region Sjælland (ID); Region Sjælland Sundhedsvidenskabelige Forskningsfond (R22-A649; R37-A1598); KEU Region Sjælland; KEU Capital Region of Denmark, Helsefonden (21-B-0422), and Den lokale forskningspulje for NSR (A1315; A1098). The funders had no role in study design, data collection and analysis, decision to publish, or preparation of the manuscript.

**Competing interests:** The authors have declared that no competing interests exist.

(MTBQ). Data on chronic conditions, healthcare utilization, and demographic information such as sex, age, and educational attainment will be collected from national registries. The outcome measures will be recorded before, during, and after implementing the intervention. Qualitative evaluation will include semi-structured interviews with healthcare professionals across various sectors as well as patients. The cost-effectiveness and Incremental Cost Effectiveness Ratio (ICER) of the CIM2 will be assessed using Diagnose Related Group rates.

## Trial registration

ClinicalTrials.gov Identifier: NCT05406193. https://clinicaltrials.gov/study/NCT05406193.

## Introduction

The increasing prevalence of multimorbidity–driven by the ageing population, and advanced diagnostic health technologies—represents a major challenge for the healthcare system and society [1–4]. Multimorbidity is commonly defined as the coexistence of two or more chronic conditions in the same person [5,6]. In the Capital Region of Denmark and Region Zealand, approximately 21% and 37% of the citizens aged 16 years and older suffer from multimorbidity [7,8]. Around 10% of people in Denmark with multimorbidity have complicated multimorbidity characterized by a severe symptom complex caused by more concomitant chronic conditions [9]. Other researchers have proposed different definitions of complicated multimorbidity such as the severity of conditions and perception of illness [10,11]. Patients with complicated multimorbidity often experience reduced health-related quality of life [12–14], high treatment burden, polypharmacy, reduced ability to work, and increased mortality [9].

In this study, we define multimorbidity as complicated when the patient has two or more of three chronic conditions (diabetes, chronic obstructive pulmonary disease, or chronic heart conditions) [15], has been hospitalised, or visited an outpatient clinic due to their chronic condition(s) during the previous year [16], and take at least five different prescription drugs assessed from the Shared Medicine Record (SMR) in general practice [17].

Care pathways for patients with complicated multimorbidity are often complex with multiple appointments, frequent ambulatory visits, hospital admission, and use of other healthcare services. The siloed organization of healthcare around single diseases, the lack of guidelines and incompatible IT systems can be a barrier to effective care for patients with complicated multimorbidity. [18].

While the above challenges described are widely recognized [19] and some consensus exists regarding key components for enhancing care [20], knowledge of the most effective organizational structure for healthcare services to deliver patient-centred, high-quality integrated care for patients with complicated multimorbidity remains limited [21,22].

In Denmark, general practice is the key organisational setting in terms of offering people with complicated multimorbidity integrated, patient-centred care. To improve care for patients with complicated multimorbidity in general practice, we developed a complex intervention care model, "A patient-centred complex intervention for multimorbidity" (CIM) [16]. The model was developed based on the Chronic Care Model [23,24], models of care for multimorbidity [21,22,25] and results from our studies in multimorbidity [15,26,27].

Based on the results from the feasibility study, we developed an improved version of the CIM model named "A patient-centred complex intervention for multimorbidity version 2"

(CIM2) [16,28]. The new CIM2 model includes improved training of healthcare professionals, strengthened identification of patients with complicated multimorbidity, adjustment of the extended consultation according to the consultation model of The Danish College of General Practitioners (DSAM), improved medical treatment, and strengthened integration of care services between healthcare organisations [16,29]. The development of the CIM2 model relies on the Medical Research Council (MRC) framework for complex interventions [30,31]. The MRC framework consists of four phases: 1) development or identification of an intervention, 2) assessment of the feasibility of the intervention and evaluation design, 3) evaluation of the interventions, and 4) impactful implementation.

The development of the extended overview consultation in the feasibility study of CIM model [16] caught the attention of the Danish Regions Organisation and The Danish Organization of General Practitioners. The collective agreement accepted in 2022 for general practice introduced a fee-for-service covering an extended overview consultation for people with complex multimorbidity [32]. The CIM2 pilot study is expected to improve and strengthen the CIM2 model, and the results will be used to inform further interventions.

## Objective

This paper describes the protocol for a pilot cluster randomised controlled trial to determine the effectiveness of a general practice-based intervention aimed at improving treatment and care for patients with complicated multimorbidity.

## Methods

### Study design

This study is a 1:1 pilot cluster RCT (Fig 1). The study will include 14 general practices, of which seven will be randomized to the intervention group and seven to the control group, providing usual care. Each practice will recruit 25 patients, amounting to 350 patients in total. At the time of recruitment, and respectively at 6 and 12 months of the intervention period, the patient's assessment of perceived patient-centred integrated care, health-related quality of life, and treatment burden will be measured in both groups. The patients in the intervention group will receive an extended overview consultation after recruitment and again after 12 months.

### Study setting

The study will take place in Region Zealand and the Capital Region of Denmark, in general practices, healthcare centres in municipalities, and hospital outpatient clinics during the 3rd quarter of 2022 through the 4th quarter of 2024. The general practices should have a minimum of 4,500 patients registered to ensure that the number of patients with complicated multimorbidity reaches the needed number of 25 patients per practice. The municipalities will be selected based on the rank of sociodemographic groups I-IV. Groups III and IV, the lowest sociodemographic groups, will be chosen as the prevalence of patients with complicated multimorbidity is expected to be higher [8]. The possible benefits of the CIM2 model are expected to be larger in patients from municipalities characterized as belonging to lower sociodemographic groups.

### Sample size

The primary outcome measure of the study is the Patients Assessment Chronic Illness Care (PACIC) questionnaire [33,34]. The difference of interest between the intervention and the control group is $\Delta = 0.36$, with a mean value of 2.86, and with an expected standard deviation

| | STUDY PERIOD | | | | | |
|---|---|---|---|---|---|---|
| | Enrolment | Allocation | Post-allocation | | | Close-out |
| TIMEPOINT | *continuously* | Baseline | EOC* | 6- month | EOC at 12-month | *12-month* |
| **ENROLMENT:** | | | | | | |
| **Eligibility screen** | X | | | | | |
| **Informed consent** | X | X | | | | |
| *Reminder* | | X | | | | |
| **INTERVENTIONS:** | | | | | | |
| *Usual care* | | | ←————————→ | | | |
| *CIM2* | | | ←————————→ | | | |
| **ASSESSMENTS:** | | | | | | |
| *PACIC* | | X | | X | | X |
| *EQ-5D-5L* | | X | | X | | X |
| *MTBQ* | | X | | X | | X |

\* Extended overview consultation (EOC)

**Fig 1. Time schedule of enrolment, interventions, and assessments on participant outcome inspired by the SPIRIT 2013 reporting guidelines.**

SD = 1, based on results from earlier studies [35,36]. The power calculations take a potential Intraclass Correlation Coefficient (ICC) of 0.1 into account. With patients being allocated to clinics with 25 patients per clinic, and a random loss to follow up on 30% [37], we base ourselves on the following random effects model for those not lost to follow up:

$$Y_i = \alpha_{Group(i)} + Z_{Clinic(i)} + \varepsilon_i, i = 1, 2 \ldots, n \tag{1}$$

so that the outcome $Y$ consists of a group-specific level $\alpha$ (intervention/control), a clinic-specific random effect $Z$, allowing for intra-clinic correlation, and an individual noise term $\varepsilon$. Simulating the above model for 10,000 times per choice of the number of clinics in each group, with individually evaluated random loss to follow up, yields a power of 78%, and 84% for 6 and 7 clinics in each group, respectively, based on a two-sided test at the level $\alpha = 0.05$. The uncertainty of these powers is found to be less the 1 percentage point, using the binomial formula. Thus, to obtain a power of 80% for detecting the specified difference $\Delta$ from the

random effects modelling, 7 clinics of 25 patients, i.e., 175 patients are needed in the intervention and control group.

As a consequence of an anticipated attrition rate (or drop-out rate) of approximately 30% due to mortality or other unforeseen circumstances, we plan to include 183 patients in each group.

## Randomization and blinding

General practices in the designated municipalities are contacted with an invitation to participate in the study. If they accept the invitation, they will receive detailed study information. The 14 participating general practices will be randomised into either the intervention group providing care as described in the CIM2 model or the control group, providing usual care. The general practices will be randomly allocated, at an allocation rate of 1:1, by a computer program. To ensure the concealment of allocation, a data manager from another organisation will provide the information on the randomization to the general practice and will be responsible for a randomization list, which will be available to the investigator. Due to the nature of the study, the general practices and the patients cannot be blinded.

## Eligibility criteria

Healthcare professionals in general practice included in the study identify eligible patients. We use the following inclusion criteria:

1. The patient has two or more of three common chronic conditions (diabetes, chronic obstructive pulmonary disease, chronic heart condition) [15].

2. The patient has been hospitalised or visited an outpatient clinic due to their chronic conditions during the previous year [16].

3. The patient takes at least five different prescription drugs assessed from SMR in the general practice [17].

Patients accepting to become part of the study get an invitation with a link to the project database in the Research Electronic Data Capture (REDCap) [38], a web-based application developed to capture data for clinical research. In the invitation from REDCap patients receive information about the project and are asked to 1) provide informed consent to participate in the study and 2) complete the baseline questionnaires. Patients will receive up to 2 reminders if they do not fill in the questionnaires. The patients will receive the questionnaires again at 6 and 12 months after inclusion and again up to 2 reminders for non-responders.

Patients, who are unable to speak Danish, have a life expectancy of less than 12 months, or cannot give their informed consent, for example people with dementia, will be excluded from the study.

## Control general practices

Healthcare professionals working in general practices allocated to the control group will receive the same information about the project concerning the recruitment of patients, collection of patient data including informed consent and patient questionnaires and the use of REDCap. Patients will receive usual care according to the Danish Health Authority's National Clinical Guideline. Any changes in the intervention period affecting the usual care or activities for the control group will be recorded.

### Intervention general practices

The elements of the CIM2 model are described in the following and is illustrated in Fig 2.

The teaching program will be developed in collaboration with The Danish College of General Practitioners. Healthcare professionals from general practice, nurses and physiotherapists from the municipalities, and healthcare professionals from the outpatient clinics participate in the training program. It will include topics on multimorbidity, project content, and methods to recruit and include patients (including informed consent). Further, the training program will offer training in the collection of patient data comprising the use of REDCap software for patient questionnaires.

The patient intervention in general practice starts with an extended consultation, lasting 45 minutes, with the GP, the patient and possibly a relative, and the nurse care coordinator. The consultation is based on the guidance for a patient-centred overview status for patients with multimorbidity published by The Danish College of General Practitioners [39]. The aim is to obtain an overview of the patient's conditions, problems, needs, and potential beneficial changes in treatment. The patient's goals, preferences, and needs are identified, and the medical treatment of the patient's conditions is prioritized. An individual care plan is developed, covering activities in the three sectors that will take place within the 12-month intervention period.

The individual care plans include 6 themes [39]: 1) Listing of important diagnoses, 2) Overview of the patient's prescription drugs, 3) Prioritising the patients' treatment goals (using shared decision-making), 4) Development of a coordinated individual care plan with telephone follow-up and future visits, 5) To lower burden of treatment, the GP will identify outpatient visits at hospitals which could be reduced or replaced by a GP visit if the patient and the specialist agree, and 6) Referral to community-based rehabilitation programs if there is clinical indication and the patient is motivated [40].

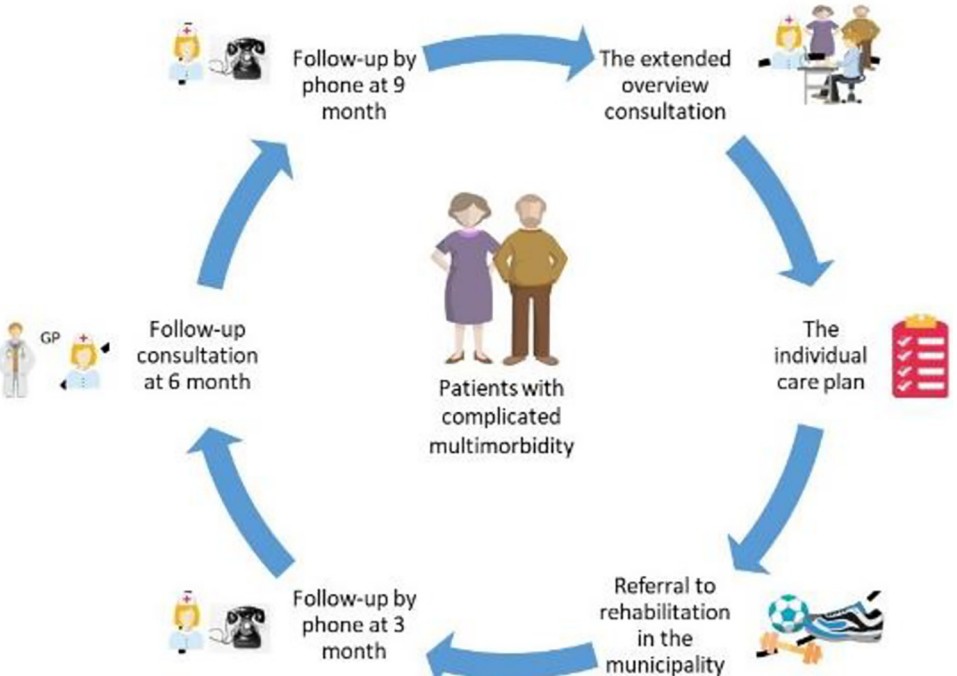

**Fig 2. An overview of the patient-centred care model for patients with complicated multimorbidity (CIM2).**

The individual care plan is printed for the patient and can be shared with the municipality and outpatient clinic based on an IT standard (MedCom). The GPs is reimbursed with 137 US dollars for the extended overview consultations.

General practice coordinates the planned patient care between general practice, the municipality, and the hospital, and follow-up on the execution of planned healthcare activities. The care coordinator function might be undertaken by the GP or the nurse in the practice. The practice plans the division of responsibilities and tasks in the project between the GP and the nurse.

The follow-up activities include telephone calls to the patient by the care coordinator at relevant time intervals according to the severity of conditions and planned activities. A second extended overview consultation takes place after 12 months.

As mentioned earlier, the individual care plan is shared electronically with the healthcare centre in the municipality and with the outpatient clinics using the national standard IT communication tool MedCom.

## Ethics

The study was notified to the Danish Data Protection Agency (protocol: REG-161-2020) and the National Committee on Health Research Ethics in Region Zealand (protocol no.: EMN-2020-37129). The study is registered at ClinicalTrials.gov, identifier: NCT05406193.

The study will be conducted in line with ethical principles for medical research as described in the Declaration of Helsinki [41]. Personal identification is encrypted, and data will be kept in accordance with the requirements of the Danish Data Protection Agency. All results will be reported in the anonymity of respondents and participating general practices.

Informed consent will be obtained from all participants before entering the study. Before signing the consent forms, participants will be informed that participation is voluntary and that they can withdraw anytime.

## Outcomes

Primary outcome is the quality of care assessed by the PACIC questionnaire. The twenty-item PACIC questionnaire with five dimensions considers whether patients receive patient-centred care measuring aspects of care most important to patients; patient activation, organisation of treatment/ decision-support, goal setting/individual adaptation, problem-solving/context, and follow-up/coordination. The PACIC questionnaire has been validated and translated into Danish [33–35].

Secondary outcomes include the patients' health-related quality of life, measured by the EuroQol-5 Domain questionnaire (EQ-5D-5L). EQ-5D-5L is a generic instrument comprised of five questions covering mobility, self-care, everyday activities, pain, discomfort, anxiety, and depression, each item on a five-point scale from no problems to extreme problems. The EQ-5D-5L has been validated and translated into Danish [11,42]. The treatment burden is measured by the Multimorbidity Treatment Burden Questionnaire (MTBQ), which has been validated and translated into Danish [43]. The utilization of health care services (number of admissions), bed days, visits to emergency departments, outpatient visits, GP visits, out-of-hour GP visits, yearly control visits in general practice and specialist visits are assessed from national register data [26]. The cost-effectiveness and Incremental Cost Effectiveness Ratio (ICER) of the CIM2 will be calculated based on Diagnose Related Group rates [44,45].

## Qualitative evaluation

The qualitative evaluation consists of semi-structured interviews with health professionals representing all three sectors, as well as patients. Semi-structured interviews will be conducted with:

1. General practitioners and nurses from the seven intervention practices.

2. Health professionals from municipalities and hospitals involved in cross-sectoral collaboration.

3. patients with a cross-sectorial course of treatment.

The semi-structured interviews with health professionals will be guided by an interview guide, which covers the following subjects: the content and relevance of the extended consultation, patient-centeredness of the care pathway, the individual care plan, referral to rehabilitation, and the content of the rehabilitation program in the community. Furthermore, semi-structured interviews with patients will explore how patients experience the extended overview consultations.

The interviews will assess the interviewees' experiences with the model, their assessment of potentials and challenges applying the model and their potential input for improving the model. One focus will be the sharing of the individual care plan and communication with patients. Interviews will be conducted once participants are familiarised with the model. All interviews will be audio-recorded, transcribed and analysed using Nvivo software.

Semi-structured interviews will be complemented with cross-sectorial focus group interviews with patients in the intervention group to assess how the CIM2, and the cross-sectorial elements associated with it, are experienced by the patients in the study. Additionally, focus group interviews will be conducted with healthcare professionals from the three sectors to gather their perspectives on the CIM2. The interviews are supported by an interview guide, which covers the following topics:

1. Teaching program for healthcare professionals: Assessing the effectiveness and comprehensiveness of the training provided to healthcare professionals prior to implementing the CIM2.

2. Structure of Extended Consultations: Evaluating the structure and organization of extended consultations in general practice settings, including their duration, format, and participant roles.

3. Integration of Care: Exploring the extent to which the CIM2 facilitates and promotes integrated care, examining collaboration and communication among healthcare providers from different sectors.

4. Information Sharing: Assessing the effectiveness of information sharing mechanisms between the three sectors, identifying potential barriers and facilitators to seamless information exchange.

5. Referral to Rehabilitation: Evaluating the process of identifying patients who require rehabilitation services and the appropriateness of referral criteria.

6. Content of Community Rehabilitation Programs: Assessing the relevance and effectiveness of community-based rehabilitation programs, ensuring alignment with patient needs and goals.

## Monitoring

As the content of the intervention does not increase risks of adverse effects on the participants, a data monitoring committee is not needed nor is a plan for stopping the intervention in case of adverse events.

## Data management

All data will be secured in a closed folder on the Region Zealand server with personal login access authorized by the primary investigator. The primary investigator has access to the full data set and the conversion key. Co-investigators will be given access when needed. The quantitative data is secured in the project database in REDCap during the intervention [38]. A data management agreement and a collaboration contract have been mutually accepted by all involved parties.

## Analyses

Descriptive data are reported as mean (SD) or median (Interquartile range) for continuous variables depending on the distribution, and as numbers and percentages for categorical variables. Descriptive statistics for demographic data will be used to describe the population in the intervention and the control groups, as well as for the populations in each of the general practices.

The primary analysis will investigate the effectiveness of the CIM2 model from the PACIC scores in the two groups. To our knowledge, no minimal important difference, which provides a 'measure of the smallest change in the patient-reported outcome of interest that patients perceive as important, either beneficial or harmful, and that would lead the patient or clinician to consider a change in management' [46], has been defined for the PACIC score. We expect a relevant change in the PACIC score to be 0.36 [35]. The analysis will be based on an intention-to-treat focus, including all participants regardless of study adherence and drop-out. To create a dataset with maximum information, missing values in the PACIC scheme will be imputed using multiple imputations if the study subject has answered at least 50% of the PACIC scheme. A linear mixed model, see Eq (1), will be used for the analysis as this model can handle the clustered observations within general practice. The variance structure will be chosen as the most suitable structure concerning the data assessed. If there is a difference between the groups at the time of randomisation, a sensitivity analysis including factors that vary between the two groups will be conducted. A comparative analysis without multiple imputations will be performed as another element of the sensitivity analysis. In addition to the intention-to-treat analysis, we will also perform a per-protocol analysis. To handle possible issues with clustered observations, the project will work with differences in the PACIC score post and before intervention. It is presumed however that this effect will be small, 0,01 to negligible in the survey population. The secondary analyses will compare scores of the EQ-5D-5L and MTBQ in the intervention group and the control group using the same methods as described.

Cost-effectiveness will be assessed by the estimation of an ICER. The ICER defines the price for an increase in quality of life (QALY) when investing in CIM2 [47]. The cost will constitute training costs, cost for the extended visits as well as the costs associated with all health care utilization in both primary and secondary care. The latter will reveal any potential effect of CIM2 on the use of health care in other sectors. QALY will be based on EQ-5D-5L measures. The time horizon will be the twelve months that the intervention lasts–hence no time discounting will be needed. The perspective will be that of the healthcare sector and the sensitivity of the ICER will be based on 95% confidence intervals estimated by probabilistic sensitivity analyses and illustrated by Cost-Effectiveness Acceptability Curves (CEACs) [48].

The qualitative interviews with patients will be analysed thematically [49] with a focus on themes related to the outcomes of the intervention, i.e., health-related quality of life and everyday experiences of living with multimorbidity. The analysis will include a focus on patients' experiences with the new consultation in general practice and will focus on the patient's experiences of integrated care.

Thematic analysis of qualitative interviews [49] with health professionals in all three sectors will identify 1) the acceptability of the intervention and 2) their experiences of possibilities and hindrances in cross-sectorial collaboration. The analyses will add to the quantitative analyses and contribute to an integral assessment of the CIM2 models' success in providing improved integrated care to patients with multimorbidity.

## Discussion

This paper describes the protocol for a pilot cluster randomised controlled trial designed to evaluate the integrated care intervention, with a focus on general practice, aimed at improving care for patients with complicated multimorbidity. The CIM2 study will contribute novel and valuable insights into multicomponent interventions targeting an expanding population.

Earlier studies have shown that GPs provided slightly fewer chronic care services than expected in practices where many patients with multimorbidity and low socioeconomic status were clustered, suggesting the inverse care law mechanisms [50]. Furthermore, providing care to patients with complicated multimorbidity in general practice in geographical areas with higher rates of patients with multimorbidity seems to increase the risk of burnout among GPs [51]. If the CIM2 model was integrated as part of usual care, it could contribute to all patients with multimorbidity were offered and received all the chronic care services they could benefit from.

Furthermore, an implementation of CIM2 could also benefit general practitioners by possibly reducing the risk of burnout [51]. A Danish study showed that a high crude rate of patient with multimorbidity increase GP's likelihood of burnout. The study concluded that general practices with a high number of patients with multimorbidity need support to prevent suboptimal care and GP burnout [49]. By implementing an extended consultation overview for patients with multimorbidity it is possible to compensate GPs economically in the existing combined fee-for-service and capitation reward system and supposedly decrease the stress levels GPs often experience caring for patients with complex care needs [50].

## Limitations

Studies on multimorbidity define multimorbidity as very heterogeneous and rarely include the severity of the conditions [5,6,52,53]. One study reported that a majority of authors using "multimorbidity" fail to provide a clear definition in their publications [52]. This makes it difficult to compare different multimorbidity studies. As mentioned earlier in this paper, we defined multimorbidity as the coexistence of two or more chronic conditions in the same person, which is one of the more common definitions [5,6]. We anticipate that this approach should facilitate comparison with previous studies.

Firstly, the study design, being a pilot cluster RCT, may have inherent limitations in terms of generalizability to broader populations. The inclusion of specific geographical regions and general practices may introduce selections bias and limit the extrapolation of findings to different healthcare settings. Additionally, the use of specific chronic conditions (diabetes, chronic obstructive pulmonary disease, chronic heart condition) as criteria for complicated multimorbidity might not cover the full spectrum of conditions affecting patients, possibly influencing the generalizability of our results.

Furthermore, the nature of the intervention, although evidence-based and informed by prior feasibility studies, may present challenges in implementation across diverse healthcare context. Factors such as variations in healthcare infrastructure, healthcare professionals, and patient populations could impact the effectiveness of the CIM2 model in different settings.

From a clinical perspective, the PACIC questionnaire, though validated, might not fully capture the complexity of patient experiences in the context of complicated multimorbidity. We use a qualitative evaluation, such as in-depth interviews, to gain a deeper understanding of patient experiences.

## Source of potential bias

A recurring challenge when conducting pilot cluster RCT studies is recruitment. There is a risk of bias during both the recruitment of general practices and patients. General practices with a high patient burden may be less likely to participate. To minimize the differences in patient burden between the general practices, we will solely recruit from municipalities with similarly low socioeconomic status. Patients recruited by their GP be more likely to participate when asked by a familiar healthcare professional whom they trust. However, there might still be a potential bias since the most vulnerable and sick patients will be less likely to participate in the study.

## Dissemination

The scientific dissemination will consist of publication in peer-reviewed scientific journals and presented at national and international conferences. The results from the study will be presented at a seminar, inviting patients, relatives, and professionals from all three sectors included in the study. Short reports on recommendations for organisational cross-sectoral and cross-disciplinary collaboration on complicated multimorbidity will be made to the ensure dissemination of key findings and implications for practice to relevant stakeholders at the regional and national levels.

## Conclusion

A successful pilot cluster RCT will yield valuable insights into recruitment strategies, how to get CIM2 implemented in general practice, municipalities, and out-patient clinics, how to improve the effectiveness and quality of the CIM2, sample size estimates for future studies and cross-sectorial collaboration. The qualitative findings will shed light on the experiences of patients and healthcare professionals with the CIM2, potentially identifying areas for improvement and informing its application in future studies.

## Supporting information

**S1 Checklist. SPIRIT 2013 checklist: Recommended items to address in a clinical trial protocol and related documents\*.**
(DOC)

**S1 Protocol.**
(PDF)

## Acknowledgments

We would like to thank general practitioners Berit Lassen and Janne Unkerskov for sharing their knowledge and experience with patients with multimorbidity, and for contributing to the discussion during the development of the CIM2.

## Author Contributions

**Conceptualization:** Sanne Lykke Lundstrøm, Nina Kamstrup-Larsen, John Sahl Andersen, Anne Frølich.

**Data curation:** Sanne Lykke Lundstrøm, Barbara Ann Barrett, Louise Marie Bidstrup Jørgensen.

**Formal analysis:** Sanne Lykke Lundstrøm, Barbara Ann Barrett, Anders Stockmarr, Anne Frølich.

**Funding acquisition:** Sanne Lykke Lundstrøm, Nina Kamstrup-Larsen, Anne Frølich.

**Investigation:** Sanne Lykke Lundstrøm, Nina Kamstrup-Larsen, Barbara Ann Barrett, Anne Frølich.

**Methodology:** Sanne Lykke Lundstrøm, Nina Kamstrup-Larsen, Barbara Ann Barrett, John Sahl Andersen, Anders Stockmarr, Anne Frølich.

**Project administration:** Sanne Lykke Lundstrøm, Barbara Ann Barrett.

**Supervision:** Sanne Lykke Lundstrøm, Barbara Ann Barrett, Anders Stockmarr, Anne Frølich.

**Visualization:** Sanne Lykke Lundstrøm, Nina Kamstrup-Larsen.

**Writing – original draft:** Sanne Lykke Lundstrøm, Nina Kamstrup-Larsen.

**Writing – review & editing:** Sanne Lykke Lundstrøm, Nina Kamstrup-Larsen, Barbara Ann Barrett, Louise Marie Bidstrup Jørgensen, Solvej Skriver Hansen, John Sahl Andersen, Bolette Friderichsen, Anders Stockmarr, Anne Frølich.

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
