## [Decision Letter · Decision Letter 0]

13 Nov 2023

PONE-D-23-28616A patient-centred care model for patients with complicated multimorbidity: Protocol for a cluster randomised trial in general practice, municipalities, and hospitalsPLOS ONE

Dear Dr. Lundstrøm,

Thank you for submitting your manuscript to PLOS ONE. After careful consideration, we feel that it has merit but does not fully meet PLOS ONE’s publication criteria as it currently stands. Therefore, we invite you to submit a revised version of the manuscript that addresses the points raised during the review process. The protocol describes a study of significance in today's global ageing population. Please consider the reviewers' feedback and queries, particularly addressing the need for more detail on the rationale for the study, to report the protocol according to the SPIRIT guideline, confusion re a pilot study or full randomised trial, and consideration of appropriate analysis tests for your outcome variables.

We look forward to receiving your revised manuscript.

Kind regards,

Kathleen Finlayson

Academic Editor

PLOS ONE

2. We note that you have indicated that data from this study are/will be made available upon request. PLOS only allows data to be available upon request if there are legal or ethical restrictions on sharing data publicly. For more information on unacceptable data access restrictions, please see http://journals.plos.org/plosone/s/data-availability#loc-unacceptable-data-access-restrictions.

a) If there are/will be ethical or legal restrictions on sharing a de-identified data set, please explain them in detail (e.g., data contain potentially sensitive information, data are owned by a third-party organization, etc.) and who has imposed them (e.g., an ethics committee). Please also provide contact information for a data access committee, ethics committee, or other institutional body to which data requests may be sent. Please note that authors, including Corresponding Authors, are not permitted to be the sole point of contact for data requests.

b) If there are no restrictions, please indicate that you will provide the minimal anonymized data set necessary to replicate your study findings as either Supporting Information files or to a stable, public repository and provide us with the relevant URLs, DOIs, or accession numbers. For a list of acceptable repositories, please see http://journals.plos.org/plosone/s/data-availability#loc-recommended-repositories.

We will update your Data Availability statement on your behalf to reflect the information you provide."

3. Please include a caption for figure 2.

4. Please upload a copy of Figure 3, to which you refer in your text on page 10. If the figure is no longer to be included as part of the submission please remove all reference to it within the text.

5. We note that Figure 2 in your submission contain copyrighted images. All PLOS content is published under the Creative Commons Attribution License (CC BY 4.0), which means that the manuscript, images, and Supporting Information files will be freely available online, and any third party is permitted to access, download, copy, distribute, and use these materials in any way, even commercially, with proper attribution. For more information, see our copyright guidelines: http://journals.plos.org/plosone/s/licenses-and-copyright.

6. Please note that the Study Protocol article type is only suitable for proposals of studies that have not yet generated results. For further information, please see https://journals.plos.org/plosone/s/submission-guidelines#loc-study-protocols

Please update your Cover Letter to indicate whether you wish to have the manuscript considered as a Study Protocol, and confirm that neither recruitment nor data collection had been completed at the time of submission (September 2023)

Reviewers' comments:

Reviewer's Responses to Questions

**Comments to the Author**

1. Does the manuscript provide a valid rationale for the proposed study, with clearly identified and justified research questions?

Reviewer #1: Partly

Reviewer #2: Partly

2. Is the protocol technically sound and planned in a manner that will lead to a meaningful outcome and allow testing the stated hypotheses?

Reviewer #1: Partly

Reviewer #2: Partly

3. Is the methodology feasible and described in sufficient detail to allow the work to be replicable?

Reviewer #1: Yes

Reviewer #2: No

4. Have the authors described where all data underlying the findings will be made available when the study is complete?

Reviewer #1: No

Reviewer #2: Yes

5. Is the manuscript presented in an intelligible fashion and written in standard English?

Reviewer #1: No

Reviewer #2: Yes

6. Review Comments to the Author

You may also provide optional suggestions and comments to authors that they might find helpful in planning their study.

Reviewer #1: Thank you for the opportunity to read and review this protocol manuscript. Overall, I find the protocol relevant and feasible.

I do however, have some suggestions for improvement.

Abstract: overall, I find the language in the abstract inconsistent and in need for revision by a native English speaker.

Especially the Introduction (in the abstract) need a re-writing regarding coherence and language. Consider not to use "often" (also in the manuscript) as it is unspecific tending to appraising.

I find the sentence "There remains limited understanding..." both clumsy and arrogant - what do you know about peoples' understanding? I know, it hasn't been the intention but nevertheless, it is the signal. CIM should be explained (or not used) in the abstract. I struggle a bit to understand the choice of the word "effects" in the aim - are you sure that this is the correct word to use here? or is it rather "impact" or "effectiveness"?

First line in the Methods (in the abstract): should it be: ".. integrated care for patients..."?

Should "patient's" be plural (patients')?

In the last 4-5 lines of the abstract you write in the past tense (were collected, were recorded, assessments involved...) - should this be the present tense or have these actions been carried out?

Introduction

Unless it is due to journal guidelines, please, gather consecutive references in one bracket.

Consider to decrease the use of "often"

The literature used to argue for a recent empirical situation (here: the organisation of healthcare, the number of persons suffering from multi-morbidity and so on) should be recent and not from e.g., 2015, 1996, 2016.

Line 5 regarding the amount of people with complicated multi-morbidity - where is this? still in Denmark?

The coherence between some of the sections in the Introduction is poor - for example the description of the secondary healthcare sector appears a bit out of context.

Reference 15 has not publication year presented in the Reference list.

Line 8 from the bottom of the section: what is a critical potential? please, clarify.

Consider moving the sentences describing that both studies run simultaneously to tie Limitations - it is a bit weird to have this information here.

Aim: this aim corroborates better with the study design and the chosen outcomes - but why does it differ substantially from the aim in the abstract? Please align.

Methods

I wonder why you do not describe that the protocol is guided by the SPIRIT - I would have recommended that but noticed by the end of the manuscript that you have... I do however, then wonder why you haven't followed the guideline? your structure differ completely from the structure recommended by the guideline.

It is stated that the study will take place in the 3rd quarter of2022 - does this mean that the study is completed? before the protocol is published?

Regarding the sample size - why do you think that 30% will dropout? Please, provide a reference for this.

Data

What is meant by "Patient outcome"? should this term be mentioned in the design..?

And why have you chosen to interview 15 patients?

Analyses

In this section, quite new terms are introduced: comparability, health economic evaluation and process evaluation - they should be introduced earlier. Please, align terms in Abstract, Methods, Data and Analyses.

References

The reference list should be scrutinized regarding Danish titles (should be presented in English in square brackets) - for example no4, 5, 25, 42 - and lack of information - for example no 15 without publication year.

Reviewer #2: Important note: This review pertains only to ‘statistical aspects’ of the study and so ‘clinical aspects’ [like medical importance, relevance of the study, ‘clinical significance and implication(s)’ of the whole study, etc.] are to be evaluated [should be assessed] separately/independently. Further please note that any ‘statistical review’ is generally done under the assumption that (such) study specific methodological [as well as execution] issues are perfectly taken care of by the investigator(s). This review is not an exception to that and so does not cover clinical aspects {however, seldom comments are made only if those issues are intimately / scientifically related & intermingle with ‘statistical aspects’ of the study}. Agreed that ‘statistical methods’ are used as just tools here, however, they are vital part of methodology [and so should be given due importance]. I look at the manuscript in/with statistical view point, other reviewer(s) look(s) at it with different angle so that in totality the review is very comprehensive. However, there should be efforts from authors side to improve (may be by taking clues from reviewer’s comments). Therefore, please do not limit the revision only (with respect) to comments made here.

COMMENTS: There are a few issues about which I have different opinion and such observations/concerns are given below (since the study is at ‘Protocol’ stage, it is possible to incorporate agreed ones):

In ‘Aim of the pilot study’ section [end of ‘Introduction’, page 5] it is stated that “This paper describes the protocol for a pilot cluster randomised controlled trial to determine the effectiveness of a general practice-based intervention aimed at improving treatment and care for patients with complicated multimorbidity”. Is this a protocol for a pilot cluster randomised controlled trial? I wonder, why (in that case) the word PILOT has not appeared in ‘title’ of the study (then)? In my opinion that is expected. Later in ‘Development of the CIM2 intervention’ section you state that “The study described in this protocol is the execution of phase 3 of the MRC framework for complex interventions”, and accordingly [according to ‘MRC framework’ described by you described in this section] a full-fledged clinical trial is expected. Then why this ‘PILOT’? Actually, the present one seems to be a full-fledged clinical trial.

In ‘Abstract - methods’ section, it is stated that “The primary outcome measure is the patient’s experience of quality of care measured by the Patient Assessment Chronic Illness Care Questionnaire (PACIC). Secondary outcomes include the patient’s health-related quality of life, measured by the EuroQol-5 Domain questionnaire (EQ-5D-5L) and the treatment burden measured by the Multimorbidity Treatment Burden Questionnaire (MTBQ).”. Note that though the measures/tools used are appropriate, all of them are likely to yield data that are in ‘ordinal’ level of measurement [and not in ratio level of measurement {as the score two times higher does not indicate presence of that parameter/phenomenon as double (for example, a Visual Analogue Scales VAS score or say ‘depression’ score)}]. Then application of suitable non-parametric (or distribution free) test(s) is/are indicated/advisable [even if distribution may be ‘Gaussian’ (also called ‘normal’)].

Agreed that there is/are no non-parametric test(s)/technique(s) available to be used as alternative in all situation(s), but should be used whenever/wherever they are available. Therefore, in short use suitable non-parametric test(s)/technique(s) while dealing with data that are in ‘ordinal’ level of measurement even if [despite that] the distribution may be ‘Gaussian’.

In ‘Analyses - Analyses of comparability’ section use of any non-parametric test/technique is/are not proposed. Instead, it is said that “A linear mixed model, will be used for the analysis”. Using mixed-model regression is not wrong at all, but note that this technique [in fact any regression technique(s) for that matter] is/are not originally developed for testing the ‘Group difference(s)’. Head-to-head comparison is expected, as through regression is an indirect/secondary/by-product testing, in my opinion. {Section ‘Analyses of comparability’ mentions that “The secondary analyses will compare scores of the EQ-5D-5L and MTBQ in the intervention group and the control group using the same methods as described.”}

Sample size estimation used is described in ‘Sample size’ section seems to be alright but is little complicated (since not usual). In this (‘Sample size’) section for few assumptions refences are given [example: The difference of interest between the intervention and the control group is Δ=0.36, with a mean value of 2.86, and with an expected standard deviation SD = 1, based on results from earlier studies [33], [34].] but the other important assumption is without any relevant refence(s) [example: The power calculations take a potential Intraclass Correlation Coefficient (ICC) of 0.1 into account.], whereas the sample size estimate depends on all [and so is very much affected due to assumed values without any relevant refence(s)]. Your sample size estimation may be correct, however, raises doubt in mind of most readers. Further, kindly confirm whether the procedure used here, takes care of (i.e., incorporates) ‘30% dropouts assumed’? Otherwise, mind you that 30% (of 175) is 53 and so needed sample size is 228.

The ‘Limitations’ section includes only defining term multimorbidity and this makeing it difficult to compare different multimorbidity studies. Does that mean {according to authors} there are none other(s)? As pointed out in ‘important note’ above “This review pertains only to ‘statistical aspects’ of the study and so ‘clinical aspects’ should be assessed separately/independently [one should carefully consider/look at the clinical implications of the study].

In my opinion, to make this article acceptable (which is quite possible), a small amount of re-vision (re-drafting) may be needed. However, please do not limit the revision only (with respect) to comments made here. More improvement is expected. The respected ‘Editor’ may consider accepting/further processing only if found ‘clinical implications’ valuable [i.e., add(s) to clinical knowledge or positively influence clinical practice]. ‘Minor revision’ is recommended.

Important note: This review pertains only to ‘statistical aspects’ of the study and so ‘clinical aspects’ [like medical importance, relevance of the study, ‘clinical significance and implication(s)’ of the whole study, etc.] are to be evaluated [should be assessed] separately/independently. Further please note that any ‘statistical review’ is generally done under the assumption that (such) study specific methodological [as well as execution] issues are perfectly taken care of by the investigator(s). This review is not an exception to that and so does not cover clinical aspects {however, seldom comments are made only if those issues are intimately / scientifically related & intermingle with ‘statistical aspects’ of the study}. Agreed that ‘statistical methods’ are used as just tools here, however, they are vital part of methodology [and so should be given due importance]. I look at the manuscript in/with statistical view point, other reviewer(s) look(s) at it with different angle so that in totality the review is very comprehensive. However, there should be efforts from authors side to improve (may be by taking clues from reviewer’s comments). Therefore, please do not limit the revision only (with respect) to comments made here.

COMMENTS: There are a few issues about which I have different opinion and such observations/concerns are given below (since the study is at ‘Protocol’ stage, it is possible to incorporate agreed ones):

In ‘Aim of the pilot study’ section [end of ‘Introduction’, page 5] it is stated that “This paper describes the protocol for a pilot cluster randomised controlled trial to determine the effectiveness of a general practice-based intervention aimed at improving treatment and care for patients with complicated multimorbidity”. Is this a protocol for a pilot cluster randomised controlled trial? I wonder, why (in that case) the word PILOT has not appeared in ‘title’ of the study (then)? In my opinion that is expected. Later in ‘Development of the CIM2 intervention’ section you state that “The study described in this protocol is the execution of phase 3 of the MRC framework for complex interventions”, and accordingly [according to ‘MRC framework’ described by you described in this section] a full-fledged clinical trial is expected. Then why this ‘PILOT’? Actually, the present one seems to be a full-fledged clinical trial.

In ‘Abstract - methods’ section, it is stated that “The primary outcome measure is the patient’s experience of quality of care measured by the Patient Assessment Chronic Illness Care Questionnaire (PACIC). Secondary outcomes include the patient’s health-related quality of life, measured by the EuroQol-5 Domain questionnaire (EQ-5D-5L) and the treatment burden measured by the Multimorbidity Treatment Burden Questionnaire (MTBQ).”. Note that though the measures/tools used are appropriate, all of them are likely to yield data that are in ‘ordinal’ level of measurement [and not in ratio level of measurement {as the score two times higher does not indicate presence of that parameter/phenomenon as double (for example, a Visual Analogue Scales VAS score or say ‘depression’ score)}]. Then application of suitable non-parametric (or distribution free) test(s) is/are indicated/advisable [even if distribution may be ‘Gaussian’ (also called ‘normal’)].

Agreed that there is/are no non-parametric test(s)/technique(s) available to be used as alternative in all situation(s), but should be used whenever/wherever they are available. Therefore, in short use suitable non-parametric test(s)/technique(s) while dealing with data that are in ‘ordinal’ level of measurement even if [despite that] the distribution may be ‘Gaussian’.

In ‘Analyses - Analyses of comparability’ section use of any non-parametric test/technique is/are not proposed. Instead, it is said that “A linear mixed model, will be used for the analysis”. Using mixed-model regression is not wrong at all, but note that this technique [in fact any regression technique(s) for that matter] is/are not originally developed for testing the ‘Group difference(s)’. Head-to-head comparison is expected, as through regression is an indirect/secondary/by-product testing, in my opinion. {Section ‘Analyses of comparability’ mentions that “The secondary analyses will compare scores of the EQ-5D-5L and MTBQ in the intervention group and the control group using the same methods as described.”}

Sample size estimation used is described in ‘Sample size’ section seems to be alright but is little complicated (since not usual). In this (‘Sample size’) section for few assumptions refences are given [example: The difference of interest between the intervention and the control group is Δ=0.36, with a mean value of 2.86, and with an expected standard deviation SD = 1, based on results from earlier studies [33], [34].] but the other important assumption is without any relevant refence(s) [example: The power calculations take a potential Intraclass Correlation Coefficient (ICC) of 0.1 into account.], whereas the sample size estimate depends on all [and so is very much affected due to assumed values without any relevant refence(s)]. Your sample size estimation may be correct, however, raises doubt in mind of most readers. Further, kindly confirm whether the procedure used here, takes care of (i.e., incorporates) ‘30% dropouts assumed’? Otherwise, mind you that 30% (of 175) is 53 and so needed sample size is 228.

The ‘Limitations’ section includes only defining term multimorbidity and this making it difficult to compare different multimorbidity studies. Does that mean {according to authors} there are none other(s)? As pointed out in ‘important note’ above “This review pertains only to ‘statistical aspects’ of the study and so ‘clinical aspects’ should be assessed separately/independently [one should carefully consider/look at the clinical implications of the study].

In my opinion, to make this article acceptable (which is quite possible), a small amount of re-vision (re-drafting) may be needed. However, please do not limit the revision only (with respect) to comments made here. More improvement is expected. The respected ‘Editor’ may consider accepting/further processing only if found ‘clinical implications’ valuable [i.e., add(s) to clinical knowledge or positively influence clinical practice]. ‘Minor revision’ is recommended.

7. PLOS authors have the option to publish the peer review history of their article (what does this mean?). If published, this will include your full peer review and any attached files.

Reviewer #1: No

Reviewer #2: No

---

## [Author Response · Author response to Decision Letter 0]

22 Dec 2023

Authors replies in red

2. We note that you have indicated that data from this study are/will be made available upon request. PLOS only allows data to be available upon request if there are legal or ethical restrictions on sharing data publicly. For more information on unacceptable data access restrictions, please see http://journals.plos.org/plosone/s/data-availability#loc-unacceptable-data-access-restrictions.

a) If there are/will be ethical or legal restrictions on sharing a de-identified data set, please explain them in detail (e.g., data contain potentially sensitive information, data are owned by a third-party organization, etc.) and who has imposed them (e.g., an ethics committee). Please also provide contact information for a data access committee, ethics committee, or other institutional body to which data requests may be sent. Please note that authors, including Corresponding Authors, are not permitted to be the sole point of contact for data requests.

b) If there are no restrictions, please indicate that you will provide the minimal anonymized data set necessary to replicate your study findings as either Supporting Information files or to a stable, public repository and provide us with the relevant URLs, DOIs, or accession numbers. For a list of acceptable repositories, please see http://journals.plos.org/plosone/s/data-availability#loc-recommended-repositories.

REPLY (also stated in the above cover letter to editor)

There are legal restrictions, but data from the pilot study may be accessed if an application to access the data is approved by government agencies. Hence, “upon request refers” to submitting a formal application to the Danish authorities. We have made the following changes:

Data Availability Statement

No datasets were generated or analysed during the current study. The European General Data Protection Regulation will safeguard the sharing of data in the pilot study. In compliance with patient privacy and Danish data protection law restrictions, the aggregated dataset utilized in the pilot study will only be accessible through Statistics Denmark, a trusted third party that manages and governs the data. Authorized scientific institutions based in Denmark can request access to this data from Statistics Denmark for individual researchers within or outside the country. Requests for accessing the data should be directed to Statistics Denmark at: https://www.dst.dk/en

3. Please include a caption for figure 2.

We have included a caption for Figure 2: Figure 2. An overview of the patient-centred care model for patients with complicated multimorbidity (CIM2).

4. Please upload a copy of Figure 3, to which you refer in your text on page 10. If the figure is no longer to be included as part of the submission please remove all reference to it within the text.

We have removed the reference as the figure is no longer to be included.

5. We note that Figure 2 in your submission contain copyrighted images. All PLOS content is published under the Creative Commons Attribution License (CC BY 4.0), which means that the manuscript, images, and Supporting Information files will be freely available online, and any third party is permitted to access, download, copy, distribute, and use these materials in any way, even commercially, with proper attribution. For more information, see our copyright guidelines: http://journals.plos.org/plosone/s/licenses-and-copyright.

We have revised the figure in Piktochart. According to the terms and policy for Piktochart “users can freely distribute visuals made using Piktochart on any medium including, TV, print, internet and mobile.” https://support.piktochart.com/article/104-using-piktochart-commercially

6. Please note that the Study Protocol article type is only suitable for proposals of studies that have not yet generated results. For further information, please see https://journals.plos.org/plosone/s/submission-guidelines#loc-study-protocols

Please update your Cover Letter to indicate whether you wish to have the manuscript considered as a Study Protocol, and confirm that neither recruitment nor data collection had been completed at the time of submission (September 2023)

REPLY (also stated in the above cover letter to editor)

We have updated the Cover Letter and indicated that we would like to have the manuscript considered as a Study Protocol. We also confirm that neither recruitment nor data collection had been completed in September 2023, when the manuscript were submitted. We are still recruiting patients and data collection will continue until spring 2025. 

Comments to the Author

Reviewers #1 Comments Authors Response 

ABSTRACT: 

Especially the Introduction (in the abstract) need a re-writing regarding coherence and language. Consider not to use "often" (also in the manuscript) as it is unspecific tending to appraising. We have re-written the abstract: 

Introduction

Current care is inadequate for patients with complicated multimorbidity, and frequently results in fragmented care. There is no widely agreed-upon optimal organisation of healthcare services for this patient group. By drawing upon existing literature and prior studies, we developed a patient-centred complex intervention for multimorbidity (CIM) and subsequently refined it into CIM version 2 (CIM2). This paper describes the study protocol for a pilot cluster randomised control trail (RCT) evaluating the effectiveness of a general practice-based intervention. 

Methods

CIM2 aims to support integrated care for patients with complicated multimorbidity. CIM2 comprises five elements: 1) Training healthcare professionals, 2) an extended overview consultation in general practice, 3) a nurse care coordinator in general practice supporting the planning of the patient trajectory, 4) follow-up care services in general practice, and 5) improving the integration of care between general practice, municipality, and hospital. The pilot cluster RCT involve 350 patients with complicated multimorbidity across 14 general practices in Region Zealand and The Capital Region of Denmark. Patients are randomly assigned to either the intervention group or the usual care group. The primary outcome measure is the patients experience of quality of care measured by the Patient Assessment Chronic Illness Care Questionnaire (PACIC). Secondary outcomes include the patient’s health-related quality of life, measured by the EuroQol-5 Domain questionnaire (EQ-5D-5L) and the treatment burden measured by the Multimorbidity Treatment Burden Questionnaire (MTBQ). Data on chronic conditions, healthcare utilization, and demographic information such as sex, age, and educational attainment will be collected from national registries. The outcome measures will be recorded before, during, and after implementing the intervention. Qualitative evaluation will include semi-structured interviews with healthcare professionals across various sectors as well as patients. The cost-effectiveness and Incremental Cost Effectiveness Ratio (ICER) of the CIM2 will be assessed using Diagnose Related Group rates.

I find the sentence "There remains limited understanding..." both clumsy and arrogant - what do you know about peoples' understanding? I know, it hasn't been the intention but nevertheless, it is the signal. We have deleted this sentence when re-writing the abstract according to the comment above. 

CIM should be explained (or not used) in the abstract. We made sure that the CIM is explained in the abstract

. I struggle a bit to understand the choice of the word "effects" in the aim - are you sure that this is the correct word to use here? or is it rather "impact" or "effectiveness"?

 We have changed the word effects to effectiveness. 

First line in the Methods (in the abstract): should it be: ".. integrated care for patients..."? Yes, thank you. We have now changed it.

Should "patient's" be plural (patients')? It is changed to plural – patients. Thank you.

In the last 4-5 lines of the abstract you write in the past tense (were collected, were recorded, assessments involved...) - should this be the present tense or have these actions been carried out? We have changed the last lines in the abstract from past tense to present continuous tense.

INTRODUCTION: 

Unless it is due to journal guidelines, please, gather consecutive references in one bracket. We have made changes in the document preferences, so consecutive references are shown in one bracket. 

Consider to decrease the use of "often" We have decreased the use of often in the introduction. 

The literature used to argue for a recent empirical situation (here: the organisation of healthcare, the number of persons suffering from multi-morbidity and so on) should be recent and not from e.g., 2015, 1996, 2016. We have updated the references.

Jerez-Roig J. Trends of multimorbidity in 15 European countries: a population-based study in community-dwelling adults aged 50 and over. BMC Public Health. 2021;21: 76–76. doi:10.1186/s12889-020-10084-x

3. Smith SM, Wallace E, O’Dowd T, Fortin M. Interventions for improving outcomes in patients with multimorbidity in primary care and community settings. Cochrane Database Syst Rev. 2021;1: CD006560–CD006560. doi:10.1002/14651858.CD006560.pub4

4. Tran PB, Kazibwe J, Nikolaidis GF, Linnosmaa I, Rijken M, van Olmen J. Costs of multimorbidity: a systematic review and meta-analyses. BMC Med. 2022;20: 1–234. doi:10.1186/s12916-022-02427-9

Line 5 regarding the amount of people with complicated multi-morbidity - where is this? still in Denmark? Yes, we have specified in the manuscript that this is in Denmark

The coherence between some of the sections in the Introduction is poor - for example the description of the secondary healthcare sector appears a bit out of context. Thank you to the reviewer for giving us the opportunity to revise the introduction to make the sections more coherent.

Reference 15 has not publication year presented in the Reference list. We have changed it to the following: 

19. Wolff JL. Prevalence, expenditures, and complications of multiple chronic conditions in the elderly. (Archives of Internal Medicine). JAMA J Am Med Assoc. 2003;289: 824-. 

Line 8 from the bottom of the section: what is a critical potential? please, clarify. We have clarified it in the manuscript: 

This has created an opportunity for a large-scale testing of the elements from the CIM2 model at a national level

Consider moving the sentences describing that both studies run simultaneously to tie Limitations - it is a bit weird to have this information here.

 After careful consideration have, we chosen to take part of this section out of the manuscript and re-written this section in the introduction: 

The development of the extended overview consultation in the feasibility study of CIM model[18] caught the attention of the Danish Regions Organisation and The Danish Organization of General Practitioners. The collective agreement accepted in 2022 for general practice introduced a fee-for-service covering an extended overview consultation for people with complex multimorbidity[34]. The CIM2 pilot study is expected to improve and strengthen the CIM2 model, and the results will be used to inform further interventions.

Aim: this aim corroborates better with the study design and the chosen outcomes - but why does it differ substantially from the aim in the abstract? Please align. We have rephrased the aim in the abstract: This paper describes the study protocol for a pilot randomised controlled trial (RCT) evaluating the effectiveness of a general practice-based intervention. 

METHODS: 

I wonder why you do not describe that the protocol is guided by the SPIRIT - I would have recommended that but noticed by the end of the manuscript that you have... I do however, then wonder why you haven't followed the guideline? your structure differ completely from the structure recommended by the guideline. Our research protocol adheres to SPIRIT guidelines. In crafting our protocol article, we meticulously selected elements from SPIRIT that were pertinent to our study's framework. To align more closely with these guidelines, we have modified certain subheadings and restructured sections of the protocol article for enhanced coherence with SPIRIT's principles. 

It is stated that the study will take place in the 3rd quarter of2022 - does this mean that the study is completed? before the protocol is published? We find this questions confusion as it is written in the manuscript that: The study will take place in Region Zealand and the Capital Region of Denmark, in general practices, healthcare centres in municipalities, and hospital outpatient clinics during the 3rd quarter of 2022 through the 4th quarter of 2024.

To clarify, the study is not completed before the protocol was submitted and will probably also not be completed before the protocol is published – depending on when it is published. The first general practices began recruiting patients in the 3rd quarter of 2022 and have continued recruiting during 2023. The extended overview consultations will continue during 2024 and maybe also during the 1st quarter of 2025. 

Regarding the sample size - why do you think that 30% will dropout? Please, provide a reference for this. We have added a reference [37]

DATA: 

What is meant by "Patient outcome"? should this term be mentioned in the design..? To not creating any confusion, we have changed it to outcome. 

And why have you chosen to interview 15 patients? In this qualitative part of the study, we have prioritized in-depth exploration of individual experiences, and we estimate that approximately 15 interviews will be appropriate to achieve a rich understanding of the phenomenon under investigation. 

ANALYSES: 

In this section, quite new terms are introduced: comparabi

---

## [Decision Letter · Decision Letter 1]

13 Mar 2024

PONE-D-23-28616R1A patient-centred care model for patients with complicated multimorbidity: Protocol for a cluster randomised trial in general practice, municipalities, and hospitalsPLOS ONE

Dear Dr. Lundstrøm,

Thank you for submitting your manuscript to PLOS ONE. After careful consideration, we feel that it has merit but does not fully meet PLOS ONE’s publication criteria as it currently stands. Therefore, we invite you to submit a revised version of the manuscript that addresses the points raised during the review process.

 ** In particular, please consider the reviewers' comments below on addressing justification for the study in the introduction and background, and the need for further details of the methods for the interviews and recruitment.**

We look forward to receiving your revised manuscript.

Kind regards,

Kathleen Finlayson

Academic Editor

PLOS ONE

Journal Requirements:

Reviewers' comments:

Reviewer's Responses to Questions

**Comments to the Author**

1. Does the manuscript provide a valid rationale for the proposed study, with clearly identified and justified research questions?

Reviewer #3: No

2. Is the protocol technically sound and planned in a manner that will lead to a meaningful outcome and allow testing the stated hypotheses?

Reviewer #3: Yes

3. Is the methodology feasible and described in sufficient detail to allow the work to be replicable?

Reviewer #3: No

4. Have the authors described where all data underlying the findings will be made available when the study is complete?

Reviewer #3: Yes

5. Is the manuscript presented in an intelligible fashion and written in standard English?

Reviewer #3: Yes

6. Review Comments to the Author

You may also provide optional suggestions and comments to authors that they might find helpful in planning their study.

Reviewer #3: Annotation Summary of PONE-D-23-28616.pdf.

Note [page 15]: Abstract

Introduction: correct spelling of trail to trial

Methods: Change to “Diagnosis- Related” rather than "Diagnose"

Note [page 16]: Introduction

To make this more readable, I suggest that you put the definitions of morbidity followed by the stats, then the definition of complicated multi morbidity, then describe the situation in Denmark - with 10% having… etc.

The first 2 paras do not read well in the current format.

Later in the introduction, please state that you are in phase 3 of the MRC framework - if this is the case.

It is important that the introduction correlates with the outcomes of the study. The outcome of your study is a patient reported measure (at least your primary outcome). Therefore, when you comment on the current state of care for multi-morbid patients, please make reference to the literature which shows that existing patient reported outcomes are poor in this group.

Your other comments in the introduction regarding the ‘silo-ing’ of care or the lack of "effective care" are probably true, but this protocol is not designed to improve any of that. The primary outcome the study measures is the effect on patient reported outcomes.

I also think the introduction can be rewritten to improve flow and readability.

Note [page 22]: Methods - Qualitative Evaluation

There is repetition here regarding the guide for semistructured interviews. Please streamline, and mention the inclusions in your guide one place only. Articulate clearly the interviews that you will conduct as part of the qualitative process. Under each type of the interviews, list the questions you will consider in the semistructured interview. In its present form, there is a series of interviews mentioned and multiple "guides to interview" descriptions.

You also need to specify whether you will interview patients / HCPs who have been randomly allocated to standard care. Do you think this will be necessary so that you can draw good comparisons to your intervention group?

Note [page 22]: Outcomes.

Please see my earlier comment regarding writing an introduction that matches to the outcomes that you are studying in the current protocol. We need to show that the current literature demonstrates that patient reported outcomes are poor with the current models of care.

Note [page 25]: Discussion

Returning to my earlier comments, it is important that you mention in the discussion that the study seeks to improve patient experiences in particular. In its current form, you have spoken, generally about "improving care for patients" or "reducing burnout for GPs “. While both are laudable aims, the purpose of your study to improve patient-reported outcomes and you must talk about this in your discussion.

Note [page 27]: Source of potential bias

Please clarify this sentence: Patients recruited by their GP be more likely to participate when asked by a familiar healthcare professional whom they trust.

Principles of good clinical practice in research suggest that recruitment should not be undertaken by practitioners involved in the treatment of patients. Therefore, it would be wrong for patients to be asked to participate by a healthcare professional whom they are familiar with or trust.

All requests for participation in a clinical studies should be made by people who are not connected to the care of the patient so as to not introduce any unwanted pressure to participate. You are right in saying this is also a potential for bias.

I am sure you are aware of this but your sentence is misleading and needs to be clarified.

If, on the other hand, you do plan to recruit patients by invitations from their own health professionals, you must change this as it is not acceptable practice.

7. PLOS authors have the option to publish the peer review history of their article (what does this mean?). If published, this will include your full peer review and any attached files.

Reviewer #3: **Yes: **Assoc Prof Rajesh Raj

---

## [Author Response · Author response to Decision Letter 1]

8 Aug 2024

Authors replies in red

Journal Requirements:

Reviewers' comments:

Reviewer's Responses to Questions

Comments to the Author

1. Does the manuscript provide a valid rationale for the proposed study, with clearly identified and justified research questions?

Reviewer #3: No

2. Is the protocol technically sound and planned in a manner that will lead to a meaningful outcome and allow testing the stated hypotheses?

Reviewer #3: Yes

3. Is the methodology feasible and described in sufficient detail to allow the work to be replicable?

Reviewer #3: No

4. Have the authors described where all data underlying the findings will be made available when the study is complete?

Reviewer #3: Yes

5. Is the manuscript presented in an intelligible fashion and written in standard English?

Reviewer #3: Yes

6. Review Comments to the Author

You may also provide optional suggestions and comments to authors that they might find helpful in planning their study.

Reviewer #3: Annotation Summary of PONE-D-23-28616.pdf.

Note [page 15]: Abstract

Introduction: correct spelling of trail to trial

We appreciate the reviewer's attentive eye in identifying the typographical error in the Introduction section of our manuscript. The incorrect spelling of "trail" has been corrected to "trial" as suggested.

Methods: Change to “Diagnosis- Related” rather than "Diagnose"

Thank you for your insightful suggestion regarding the terminology used in the Methods section of our manuscript. We have made the recommended change to ensure the manuscript's precision and readability

Note [page 16]: Introduction

To make this more readable, I suggest that you put the definitions of morbidity followed by the stats, then the definition of complicated multi morbidity, then describe the situation in Denmark - with 10% having… etc.

The first 2 paras do not read well in the current format.

Later in the introduction, please state that you are in phase 3 of the MRC framework - if this is the case.

It is important that the introduction correlates with the outcomes of the study. The outcome of your study is a patient reported measure (at least your primary outcome). Therefore, when you comment on the current state of care for multi-morbid patients, please make reference to the literature which shows that existing patient reported outcomes are poor in this group.

Your other comments in the introduction regarding the ‘silo-ing’ of care or the lack of "effective care" are probably true, but this protocol is not designed to improve any of that. The primary outcome the study measures is the effect on patient reported outcomes.

I also think the introduction can be rewritten to improve flow and readability.

Thank you for your detailed feedback and constructive suggestions for improving the introduction of our manuscript. Your input is invaluable in helping us refine our presentation and ensuring that it accurately reflects the study's scope and objectives. Here is the revised introduction that incorporates your recommendations:

Introduction

Multimorbidity is commonly defined as the coexistence of two or more chronic conditions in the same person [1,2]. The increasing prevalence of multimorbidity is largely driven by the ageing population, and advanced diagnostic health technologies, presenting a major challenge for the healthcare system and society [3–6]. In the Capital Region of Denmark and Region Zealand, approximately 21% and 37% of the citizens aged 16 years and older suffer from multimorbidity [7,8]. 

Complicated multimorbidity, a subset of the diagnosis multimorbidity is characterized by a severe symptom complex caused by more concomitant chronic conditions, and affects about 10% of people with multimorbidity in Denmark [9]. Definitions of complicated multimorbidity vary, encompassing aspects such as the severity of conditions and patients’ perceptions of their illness [10,11]. 

Due to the single disease focus, current care models for patients with multimorbidity are often fragmented and fail to provide integrated, patient-centered care. This is particularly evident from patient-reported outcomes, which consistently reveal dissatisfaction among this group. Studies have highlighted that patients with multimorbidity frequently experience poor health related quality of life, high treatment burdens, and a sense of being underserved by the healthcare systems [12–14]. These outcomes underscore the urgent need for revised care models that prioritize the patient experience and improve the overall management of multiple chronic conditions.

In this study, we define multimorbidity as complicated when the patient has two or more of three specific chronic conditions (diabetes, chronic obstructive pulmonary disease, or chronic heart conditions) [15], has been in contact with the hospital during the previous year or visited an outpatient clinic due to their chronic condition(s) during the previous year [16], and take at least five different prescription drugs assessed from the Shared Medicine Record (SMR) [17]. 

Care pathways for patients with complicated multimorbidity are often complex with multiple appointments, frequent ambulatory visits, hospital admission, and use of other healthcare services. The siloed organization of healthcare around single diseases, the lack of guidelines for multimorbidity and incompatible IT systems can be a barrier to effective care for those patients [18].

While the above challenges described are widely recognized [19] and some consensus exists regarding key components for enhancing care [20], knowledge of the most effective organizational structure for healthcare services to deliver patient-centred, high-quality integrated care for patients with complicated multimorbidity remains limited [14,21,22].

In Denmark, general practice is the key organisational setting in terms of offering people with complicated multimorbidity integrated, patient-centred care. To improve care for patients with complicated multimorbidity in general practice, we developed a complex intervention care model, “A patient-centred complex intervention for multimorbidity” (CIM) [16]. The model was developed based on the Chronic Care Model [23,24], models of care for multimorbidity [21,22,25] and results from our studies in multimorbidity [15,26,27]. 

Based on the results from the feasibility study, we developed an improved version of the CIM model named “A patient-centred complex intervention for multimorbidity version 2” (CIM2) [16,28]. The new CIM2 model includes improved training of healthcare professionals, strengthened identification of patients with complicated multimorbidity, adjustment of the extended consultation according to the consultation model of The Danish College of General Practitioners (DSAM), improved medical treatment, and strengthened integration of care services between healthcare organisations [16,29]. The development of the CIM2 model relies on the Medical Research Council (MRC) framework for complex interventions [30,31], confirming that this study is in phase 3, focusing on the evaluation of interventions. 

The development of the extended overview consultation in the feasibility study of CIM model [16] caught the attention of the Danish Regions Organisation and The Danish Organization of General Practitioners. The collective agreement accepted in 2022 for general practice introduced on a trial basis a fee-for-service covering an extended overview consultation for people with complex multimorbidity[32]. The CIM2 pilot study is expected to improve and strengthen the CIM2 model, and the results will be used to inform further interventions.

Note [page 22]: Methods - Qualitative Evaluation

There is repetition here regarding the guide for semistructured interviews. Please streamline, and mention the inclusions in your guide one place only. Articulate clearly the interviews that you will conduct as part of the qualitative process. Under each type of the interviews, list the questions you will consider in the semistructured interview. In its present form, there is a series of interviews mentioned and multiple "guides to interview" descriptions.

You also need to specify whether you will interview patients / HCPs who have been randomly allocated to standard care. Do you think this will be necessary so that you can draw good comparisons to your intervention group?

Thank you for your feedback regarding the focus of our qualitative evaluation. Based on your suggestion, we have decided to concentrate the qualitative analysis on gathering experiences from participants who have directly engaged with the CIM2 model. This approach is aimed at deeply understanding how the model functions in a real-world setting and assessing its practical viability. Therefore, we will not include participants from the standard care group in this part of the study, as their input would not contribute to our primary objective of evaluating the model's direct impacts. This method ensures that our qualitative data is highly relevant and focused on the intervention's effects.

Here is the revised qualitative evaluation that incorporates your recommendations:

The qualitative evaluation employs two primary methods – semi-structed interviews and focus group interviews. Semi-structured interviews will be conducted with general practitioners and nurses from the seven intervention practices and with patients with a cross-sectorial course of treatment. Focus group interviews interview will be conducted with health professionals representing all three sectors. The interviews are designed to capture comprehensive insight into the interviewees’ experiences with CIM2, their assessment of the potential and challenges of using the model, and their potential input for improving the model. The semi-structured interviews are guided by interview guides.

 For the healthcare professionals, the following experiences/topics are queried: Recruitment of patients, the content and relevance of the extended consultation, medical benefits of the extended overview consultation, their perception of patient benefits, referral to municipal rehabilitation and/or to hospital, individual treatment plan, cross-sectoral education of healthcare professionals, and suggestions for improvements. 

For the patients, the following topics are queried: Everyday life and networks, experience of the overview consultation including communication and patient-centeredness, individual treatment plan, experiences with cross-sectorial course of treatment and suggestions for improvement.

The focus group interview guide will address the following topics: The cross-sectorial teaching program for healthcare professionals, integration of care via CIM2, information sharing, referral to rehabilitation and the content of community rehabilitation programs.

All interviews will be recorded, transcribed, and analyzed using Nvivo software.

Note [page 22]: Outcomes.

Please see my earlier comment regarding writing an introduction that matches to the outcomes that you are studying in the current protocol. We need to show that the current literature demonstrates that patient reported outcomes are poor with the current models of care.

This has now been incorporated in the revised introduction in the above comment.

Note [page 25]: Discussion

Returning to my earlier comments, it is important that you mention in the discussion that the study seeks to improve patient experiences in particular. In its current form, you have spoken, generally about "improving care for patients" or "reducing burnout for GPs “. While both are laudable aims, the purpose of your study to improve patient-reported outcomes and you must talk about this in your discussion.

Thank you for your feedback. We appreciate the opportunity to revise our discussion to better align with the focus. We have added the following paragraph to the discussion:

The study’s primary outcome measure is the patients’ experience of quality of care, assessed using the PACIC, with secondary outcomes including health-related quality of life and treatment burden. By evaluating patient-reported outcomes and conducting qualitative assessment through interviews with patients and healthcare professionals, the study aims to assess the effectiveness of the CIM2 model in improving care for patients with complicated multimorbidity. 

Note [page 27]: Source of potential bias

Please clarify this sentence: Patients recruited by their GP be more likely to participate when asked by a familiar healthcare professional whom they trust.

Principles of good clinical practice in research suggest that recruitment should not be undertaken by practitioners involved in the treatment of patients. Therefore, it would be wrong for patients to be asked to participate by a healthcare professional whom they are familiar with or trust.

All requests for participation in a clinical studies should be made by people who are not connected to the care of the patient so as to not introduce any unwanted pressure to participate. You are right in saying this is also a potential for bias.

I am sure you are aware of this but your sentence is misleading and needs to be clarifie

---

## [Editor Report · Decision Letter 2]

6 Sep 2024

A patient-centred care model for patients with complicated multimorbidity: Protocol for a cluster randomised trial in general practice, municipalities, and hospitals

PONE-D-23-28616R2

Dear Dr. Lundstrøm,

We’re pleased to inform you that your manuscript has been judged scientifically suitable for publication and will be formally accepted for publication once it meets all outstanding technical requirements.

Kind regards,

Kathleen Finlayson

Academic Editor

PLOS ONE

Additional Editor Comments:

Thank you for the revised article addressing the reviewers' feedback. There are just a few areas where inconsistency needs amending, where you have addressed feedback in one spot however not in others. Specifically,

- the aim in your abstract should include your primary outcome, i.e., effectiveness of xxx on ?outcome. The objective on p.4 does not match this aim, as 'treatment' is not your primary outcome, this should be patients' experience of quality care

- should economic outcomes be included in your secondary outcomes?

- in the sources of bias section, consider the reviewer's comments about the conflict of interest involved in relying on the patients' GPs to recruit, there is an ethical issue

- double check the grammar throughout, e.g. the pilot RCT WILL involve ....; DRGs should be Diagnosis (not diagnose as in abstract) etc.

---

## [Editor Report · Acceptance letter]

15 Sep 2024

PONE-D-23-28616R2 

PLOS ONE

Dear Dr. Lundstrøm, 

I'm pleased to inform you that your manuscript has been deemed suitable for publication in PLOS ONE. Congratulations! Your manuscript is now being handed over to our production team.

Kind regards, 

on behalf of

Dr. Kathleen Finlayson 

Academic Editor

PLOS ONE